# Nano-Needle Boron-Doped Diamond Film with High Electrochemical Performance of Detecting Lead Ions

**DOI:** 10.3390/ma16216986

**Published:** 2023-10-31

**Authors:** Xiaoxi Yuan, Mingchao Yang, Xu Wang, Yongfu Zhu, Feng Yang

**Affiliations:** 1Key Laboratory of Automobile Materials, Ministry of Education, School of Materials Science and Engineering, Jilin University, Changchun 130022, China; xxyuan@jlenu.edu.cn (X.Y.);; 2Institute for Interdisciplinary Quantum Information Technology, Jilin Engineering Normal University, Changchun 130052, China; 3Department of Physics, Hebei Normal University of Science and Technology, Qinhuangdao 066000, China; yangmc17@mails.jlu.edu.cn

**Keywords:** boron-doped diamond, nano-needle, electrochemical, detect, lead ion

## Abstract

Nano-needle boron-doped diamond (NNBDD) films increase their performance when used as electrodes in the determination of Pb^2+^. We develop a simple and economical route to produce NNBDD based on the investigation of the diamond growth mode and the ratio of diamond to non-diamond carbon without involving any templates. An enhancement in surface area is achievable for NNBDD film. The NNBDD electrodes are characterized through scanning electron microscopy, Raman spectroscopy, X-ray diffraction, cyclic voltammetry, electrochemical impedance spectroscopy, and differential pulse anodic stripping voltammetry (DPASV). Furthermore, we use a finite-element numerical method to research the prospects of tip-enhanced electric fields for sensitive detection at low Pb^2+^ concentrations. The NNBDD exhibits significant advantages and great electrical conductivity and is applied to detect trace Pb^2+^ through DPASV. Under pre-deposition accumulation conditions, a wide linear range from 1 to 80 µgL^−1^ is achieved. A superior detection limit of 0.32 µgL^−1^ is achieved for Pb^2+^, which indicates great potential for the sensitive detection of heavy metal ions.

## 1. Introduction

Boron-doped diamond (BDD) is a promising electrode because of its outstanding electrochemical properties [1,2,3,4,5] (such as high conductivity, wide electrochemical potential window [6], chemical inertness [7], biocompatibility [8], low-noise characteristic, and high corrosion resistance [9]) when compared to other forms of carbon [10]. The application potential of BDD could be significantly improved by developing a simple and reproducible approach to increase the specific surface area of the electrode. Generally, a large specific surface area of the sensor provides more active sites for the reaction to improve its sensitivity in electroanalysis [11,12]. Therefore, nanostructured BDD has extremely broad application prospects in electrochemical sensors [13,14].

Several methods for constructing nanostructured diamond electrodes have been reported in the literature recently. These approaches can be roughly divided into top-down and bottom-up means [15]. The top-down approaches mainly refer to obtaining a three-dimensional nanostructure by etching the diamond surfaces with a mask or an external source (plasma etching [16], catalytic etching [17], thermal etching [18], etc.). However, the economic feasibility of these preparation methods regarding the complicated pretreatment or mask removal in the process hinders subsequent extensive applications. In contrast, the bottom-up methods comprise a porous or nanostructured template as the substrate material and the deposition of diamond film on the substrate. However, the substrate material must meet various stringent requirements such as high porosity, high mechanical strength, good thermal conductivity, and resistance to high temperature for chemical vapor deposition. The requirements for the preparation methods mentioned above greatly restrict the application of preparing electrodes.

Herein, we explore nano-needle boron-doped diamond (NNBDD) electrodes while ignoring the harsh requirements without masks. Based on the investigation of the diamond growth mode and the ratio of diamond to non-diamond carbon (NDC), it is possible to obtain NNBDD by removing the NDC. We aim to use the NNBDD electrode for the electrochemical detection of Pb^2+^. It is known to all that the existence of trace Pb^2+^ in drinking water could cause significant damage to human health such as kidney or liver diseases, nervous system dysfunction, and anemia, due to its high toxicity, nonbiodegradable, and bio-accumulation features [19,20,21]. The highest level of Pb^2+^ allowed in drinking water is 6 µgL^−1^ [22]. However, a better detection limit is advantageous for identification for early preventions and interventions for the cumulative release process.

Therefore, it is essential to develop a convenient, sensitive, and reliable technique for monitoring the Pb^2+^. Several typical techniques, such as fluorescence spectrometry [23], atomic absorption spectroscopy (AAS) [24], atomic emission spectroscopy (AES) [25], inductively coupled plasma mass spectrometry (ICP-MS) [26], and inductively coupled plasma optical emission spectrometry (ICP-OES) [27], have been developed for the determination of trace Pb^2+^ [28]. Compared to the conventional techniques above, differential pulse anodic stripping voltammetry (DPASV) is an in situ electrochemical approach for the measurement of trace Pb^2+^ owing to its powerful advantages of high sensitivity, rapid analysis, and instrumental portability [29]. Nanoplatelet-like or needle-like diamond has been successfully promoted in the determination of various substances [30,31], due to its high electrocatalytic activity. Therefore, NNBDD holds fascinating potential for exhibiting great performance in the detection of Pb^2+^.

In this research, a simple and economical route is demonstrated to fabricate NNBDD without involving any templates. An enhancement in surface area is achievable for NNBDD film. To better investigate the growth mechanism of NNBDD, the surface is characterized by using scanning electron microscope (SEM), Raman, and X-ray diffraction (XRD) measurements. The NNBDD exhibits a large specific surface and great electrical conductivity and is applied for the determination of trace Pb^2+^ through DPASV. Furthermore, we use a finite-element numerical method to research the prospects of tip-enhanced field intensification. Overall, it presents significant advantages in the determination of trace Pb^2+^ based on an NNBDD electrode compared with previously reported electrodes. A wide linear range from 1 to 80 µgL^−1^ and a low detection limit of 0.32 µgL^−1^ are achieved for detecting Pb^2+^, indicating the great potential of NNBDD toward the sensitive detection of heavy metal ions.

## 2. Materials and Methods

### 2.1. Materials

Pb(NO_3_)_2_ (99.9%) powders were used as reagents and obtained from Sigma-Aldrich, Shanghai, China. An amount of 0.1 M acetate buffer prepared with sodium acetate and glacial acetic acid was used throughout the determination experiment. The other chemical reagents such as potassium ferricyanide (II), potassium ferrocyanide trihydrate (III), and potassium chloride (Sigma-Aldrich, Shanghai, China) were analytical-grade and used without further purification. Ultrapure water (18.2 MΩ·cm) was used to prepare aqueous solutions for all experiments.

### 2.2. Preparation of the NNBDD Electrode

The NNBDD films were prepared on P-type Si substrates, using a microwave plasma chemical vapor deposition system at 2.45 GHz. Before the deposition of diamond films, the mirror-polished substrates were scratched with nanodiamond powders (about 5 nm) for 30 min on abrasive paper, and then ultrasonicated in an acetone solution with nano-diamond powder for 60 min to form nucleation sites. The substrates were ultrasonically cleaned with acetone, ethanol, and purified water for 10 min, and then dried with nitrogen. The nanodiamond seeds were mostly found in scratches. A small part of the diamond powder remained in the smooth area due to the impact of nanodiamonds on the silicon wafer during ultrasonic action (Appendix A). The scratches and residual nanodiamonds on the substrates contributed to diamond nucleation during the growth. The reaction gas sources included methane (CH_4_) and hydrogen (H_2_). The liquid trimethyl borate (B(OCH_3_)_3_) was carried by bubbling H_2_ gas as the boron source [32]. Firstly, we used a CH_4_/H_2_/B flow rate of 20/200/2 sccm to create the composite films containing NNBDD and NDC for 6 h. Secondly, the NNBDD/NDC composite in a porcelain boat was annealed to remove the NDC in a quartz tube at 800 °C for 15 min in the air. Lastly, the NNBDD was fabricated by rapidly pulling out the porcelain boat, which can quickly cool down to room temperature within 60 s. The diamond and NDC were etched at the same time at a high temperature (800 °C), while the NDC was etched faster. The NNBDD was retained after rapid cooling.

### 2.3. Apparatus

SEM (JSM-6480LV, Akishima, Japan) was conducted to characterize the surface morphology of the NNBDD films. The carbon phase composition was investigated through Raman spectroscopy (Renishaw inVia Raman microscope, London, UK), using laser excitation with 532 nm. The analysis of crystal structure was performed through XRD (RigakuD/MAX-RA, Akishima, Japan) using CuKα radiation at a wavelength of 1.5418 Å. Optical emission spectroscopy (OES, by Ocean Optics USB4000, Shanghai, China) was applied to measure certain species (CH, C_2_) produced through the growth of the composite (NNBDD and NDC). The surface bonds and the surface chemical states could be recorded using X-ray photoelectron spectroscopy (XPS, VG ESCALAB MK II, Dewsbury, Britain). All electrochemical characterizations of the NNBDD film were carried out on the Electrochemical Workstation (CHI 760E, Shanghai, China).

### 2.4. Electrochemical Measurements

In a three-electrode system, platinum wire and a saturated calomel electrode served as the counter and reference electrodes, respectively. The NNBDD electrode served as the working electrode. The geometric area of the NNBDD electrode was 0.10 cm^2^. The electrochemical impedance spectroscopy (EIS) spectra were measured in a solution containing 5 mM Fe(CN)_6_^3−/4−^ and 0.1 M KCl. The electrochemical measurements carried out in an acetate buffer of the NNBDD electrode were investigated by using cyclic voltammetry (CV) and DPASV using an Electrochemical Workstation at room temperature.

### 2.5. COMSOL Multiphysics Simulations

With the free electrons on the NNBDD electrodes, the electric field near the electrode was simulated using COMSOL Multiphysics 5.6. The two-dimensional numerical model was used in this study. The tip radius of the COMSOL structure was 5, 50, and 100 nm. To improve the sensitivity and electric field distribution of electrodes, three different electrode shapes were proposed in this study. The conductivity of acetate buffer was  2 *×* 10^−6^ S/m. The formula of electric field E was E = −∇V, which was calculated as the negative gradient of potential V.

## 3. Results and Discussion

### 3.1. Morphology and Structure of NNBDD Films

The SEM images (Figure 1a) show the typical cauliflower-like morphology of the nano-needle boron-doped diamond/non-diamond carbon (NNBDD/NDC) composite film. The size of cauliflower clusters is 2–3 mm and obvious diamond micro-grain particles are not visible when using the high concentration of CH_4_ with a CH_4_/H_2_/B flow rate of 20/200/2 sccm. The NNBDD film is fabricated by etching the NDC phase by annealing in a quartz tube at 800 °C for 15 min in the air from the composite. The remaining cauliflower clusters of NNBDD are clearly shown in Figure 1b. The nanoneedle-like BDD grains are randomly oriented, possessing a length of 50–250 nm, and the tip curvature of needle-shaped diamonds is a few nanometers (the inset at high orders of magnitude). In comparison to previous ways of creating NNBDD films, this method is simpler, low-cost, and efficient due to the controllable process, the lack of use of complex templates, and the absence of expensive etching equipment.

The OES result of the growth stage of the NNBDD/NDC composite film is shown in Figure 2. The emission lines from atomic hydrogen of H_α_ (656 nm) and H_β_ (486 nm), molecular hydrogen of H_2_ (580 nm), carbonaceous CH (432 and 766 nm) bands, and C_2_ (516 nm) are presented. When the concentration of CH_4_ is low, only carbonaceous CH bands are present, resulting in a high-quality growth of BDD. However, when the concentration of CH_4_ is high, a large amount of carbonaceous C_2_ bands is generated, leading to an imperfect growth of BDD with NDC. We use this method of mixed growth of diamond and NDC to obtain the composite. A large amount of C_2_ forms secondary nucleation during the diamond growth compared with NDC. This is why the BDD grains cannot grow. Then, the NDC is removed from the composite to obtain the NNBDD.

The corresponding Raman spectrum (blue line in Figure 3) supports the fact that the quality of the NNBDD/NDC composite film is poor. A broad band centered at 1550^−1^ cm appears to be related to the NDC when using a high concentration of CH_4_ during the process of diamond growth, and the characteristic diamond peak is weak at 1332 cm^−1^. The asymmetry of the phonon band is related to the Fano effect. The two broad bands centered at 500 cm^−1^ and 1200 cm^−1^ agree with the two maxima of the phonon density of states of diamond for boron doping [33]. After an annealing treatment of 15 min at 800 °C, the NDC between the composites is etched away, and the NNBDD is formed. The Raman spectrum of NNBDD comprises a prominent diamond peak at 1332 cm^−1^ after annealing treatment and the broad band about the NDC centered at 1550 cm^−1^ disappears (red line in Figure 3). Upon annealing the sample in the air at 800 °C, a major fraction of the hydrogen termination is removed from the NNBDD surfaces, and the profile asymmetry is related to Fano interference rather than the hydrogen termination [34]. It is worth pointing out that the Fano interference could be used to qualitatively study the corresponding boron concentration distribution in diamond films. The boron concentration [B] can be quantitatively estimated by fitting a band near 500 cm^−1^ according to the following imprecise empirical law, [B] (cm^−3^) = 8.44 × 10^30^ × exp[−0.048 W (cm^−1^)] [35], where W is the wavenumber of the Lorentzian component of the 505 cm^−1^ broad peak [36]. The calculated boron concentration of the NNBDD is about 3.19 × 10^20^ cm^−3^.

The XRD spectra of the NNBDD/NDC composite before annealing and NNBDD after annealing are presented in Figure 4. The peaks at 43.8°, 75.4°, 91.6°, and 119.4° correspond to the (111), (220), (311), and (400) diamond diffraction modes. For the polycrystalline NNBDD film, the dominant (111) and (220) peaks mean that the film mainly consists of (111) and (110) grains. The wide peaks between 20° and 40° in the XRD spectra could correspond to amorphous carbon. There is a significant amount of amorphous carbon in the NNBDD/NDC composite before annealing. The broad peak of amorphous carbon weakens after annealing. The residual broad peak might be attributed to some grain boundaries containing amorphous carbon components in NNBDD.

### 3.2. Electrochemical Performance

The CV tests of the NNBDD/NDC composite electrode and the NNBDD electrode in Figure 5a prove that the estimated surface area of the NNBDD electrode is 7 times that of the NNBDD/NDC composite electrode, meaning that the NNBDD structure provides more active sites for electrochemical detection. Figure 5b shows the EIS spectra that present the electron transfer kinetics of the NNBDD/NDC composite electrode and NNBDD electrode. Compared with the NNBDD/NDC composite electrode, the NNBDD has a lower charge transfer resistance, demonstrating that NNBDD has a faster charge transfer rate at the interface of the electrode and solution. The large specific surface area and better charge transfer ability of NNBDD would be beneficial for improving its electrochemical detection performance.

### 3.3. Electrochemical Characterization for Detecting Pb^2+^

Under the pre-deposition accumulation conditions (pH = 5.0, −0.8 V, and 270 s), DPASV analysis is carried out on the NNBDD electrode using Pb^2+^ standard solution with different concentrations. Figure 6a shows a selection of typical DPASV curves for Pb^2+^ from 1 to 80 µgL^−1^. Within the concentration range, the stripping peak slightly shifts to a more positive potential with increasing concentration. For example, the peak DPASV potentials for 20, 40, 60, and 80 µgL^−1^ Pb^2+^ are −0.56, −0.55, −0.54, and −0.53 V, respectively. This is consistent with previous studies where the phenomenon is owed to the equilibrium reduction potential of M^n+^/M increasing and the enlarged size of metal particles deposited on the NNBDD electrode surface with the increased ion concentration [37,38]. The stripping peaks obtained on the NNBDD electrode are asymmetric, which are attributed to the heterogeneously active electrode where the different graphically orientated grains are characterized by the different electrical conductivity and surface structure.

The calibration plot shown in Figure 6b is linear with good correlation coefficients in the range from 1 to 80 µgL^−1^ for Pb^2+^. Based on the response of three times the standard deviation of the zero-pose response, the calculated limit of detection is 0.32 µgL^−1^. These results verify that the NNBDD film is potentially utilized for the application of Pb^2+^ detection. Compared to the reported BDD electrodes detecting Pb^2+^ listed in Table 1, the NNBDD electrode has a good linear range and a greater detection limit in the literature.

For comparison, the chronoamperometry technique is employed for testing the linear response of Pb^2+^ from 1 to 80 µgL^−1^. The current/time response of the NNBDD is presented in Appendix A. High noise is seen during the measurements, increasing with the Pb^2+^ concentration. Furthermore, low concentrations of Pb^2+^ (1–9 µgL^−1^) are not detectable. To obtain high signal values (mainly at low concentration values) and a wide linear range, the DPASV method shows a much better detecting performance. Thus, the DPASV method is chosen for the investigation of the behavior of the NNBDD electrode.

### 3.4. Electric Field Simulated using COMSOL Multiphysics

Furthermore, to gain deep insight into the stripping process of Pb^2+^ deposits, we simulate the electric field distribution within the vicinity of the NNBDD electrode using COMSOL Multiphysics. To estimate the quantitative impact of the high-curvature structure on the electric field, the electric field directly adjacent to the electrode surface is mapped. Cones with rounded tips are used to represent sharp tips of NNBDD immersed in the electrolyte, and the electric field distribution around the electrode with a tip radius of 5, 50, and 100 nm is given in Figure 7. The maximum electric field around the electrode tip increases by 4.8 times as the tip radius of the electrode sharpens from 100 nm to 5 nm. It shows that the high-curvature structure can significantly enhance the electric field, which might attract Pb^2+^ near the NNBDD tip at low Pb^2+^ concentrations. The simulation results above offer guidelines that an enhanced electric field at high-curvature sites would facilitate the precipitation of Pb^2+^ and detect Pb^2+^ at low concentrations through DPASV.

### 3.5. Reproducibility and Selectivity of NNBDD

Reproducibility is an important indicator that reflects the precision of the electrode. Improving electrode reproducibility is critical to encouraging its use in online monitoring [43]. The relative standard deviation (RSD) value of the stripping peak current value from six repetitive experiments is used to evaluate the NNBDD electrode reproduction after repeated detection of 80 µgL^−1^ of Pb^2+^. A steady potential of +0.5 V is supplied to eliminate the residual from the NNBDD electrode for 600 s following each measurement. The stripping peak current value of Pb^2+^ slightly reduces as the detection number increases without the shift in stripping peak potential. The RSD value of the NNBDD electrode is calculated to be 3.8%, indicating that NNBDD has a satisfactory reproducible precision.

Selectivity indicates the electrode’s anti-interference capacity for performance in complicated water environments. Several interfering ions including Cd^2+^, Zn^2+^, Ca^2+^, Cu^2+^, Mg^2+^, Na^+^, Al^3+^, and Fe^3+^ are individually added to a standard solution of Pb^2+^ with the ten-times-higher concentration of Pb^2+^. As demonstrated in Figure 8, the signal of Pb^2+^ slightly changes when the ions of Cd^2+^, Zn^2+^, Ca^2+^, Mg^2+^, Na^+^, Al^3+^, and Fe^3+^ are added. This means that the NNBDD electrode has better anti-interference properties for the seven ions above, while the addition of Cu^2+^ produces a significant drop in the signal of detecting Pb^2+^. This might be attributed to the fact that Cu^2+^ may have a significant competitive interaction with Pb^2+^. Thus, Cu^2+^ should be pretreated in the detection of Pb^2+^ by the NNBDD.

The XRD exhibits crystallinity states of the NNBDD without any change before and after electrochemical sensing tests (Appendix A). For better insight into the chemical states of the NNBDD after electrochemical sensing tests, XPS measurements are carried out and are shown in Figure 9. The appearance of peaks at 191.7 eV, 284.6 eV, 532.8 eV, and 975.0 eV represent the binding energies of B 1 s, C 1 s, O 1 s, and O KLL, respectively [45]. This indicates that the content of the O element on the NNBDD surface increases from 7.39% to 14.24% after electrochemical sensing tests, which proves that the chemical states of the NNBDD change. However, the reproducibility and selectivity of NNBDD tests mentioned above imply that a slight variation in surface chemical state does not have a significant impact on electrode performance.

## 4. Conclusions

In this study, we develop a highly electroactive NNBDD through simple and economical annealing treatment. The NNBDD electrode sensor exhibits good linearity from 1 to 80 µgL^−1^ for detecting Pb^2+^ and realizes a good detection limit of 0.32 µgL^−1^. It is demonstrated that numerous nanoneedles provide a high specific surface area for more electrochemical active points and the sharp-tip-enhancing electric field intensification might attract Pb^2+^ to increase the local concentration of Pb^2+^ near the NNBDD electrode. This would play an important role in realizing the excellent detection limit. The NNBDD sensor shows high reproducibility and selectivity for detecting trace Pb^2+^. This sensor could be potentially utilized for the sensitive detection of other heavy metal ions.

## Figures and Tables

**Figure 1 materials-16-06986-f001:**
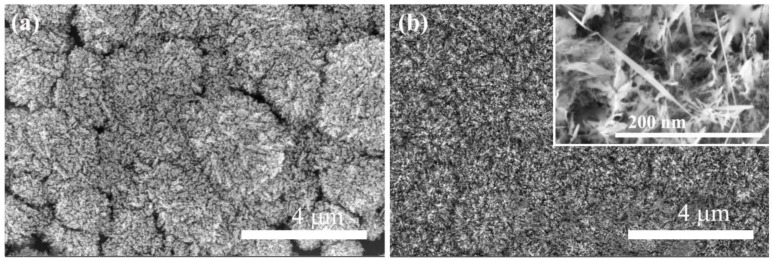
SEM images of (**a**) NNBDD/NDC composite film deposited using CH_4_/H_2_/B flow rate of 20/200/2 sccm and (**b**) NNBDD film fabricated by etching the NDC phase (annealing in a quartz tube at 800 °C for 15 min in the air) from composite. Inset graph is the image obtained at high magnification.

**Figure 2 materials-16-06986-f002:**
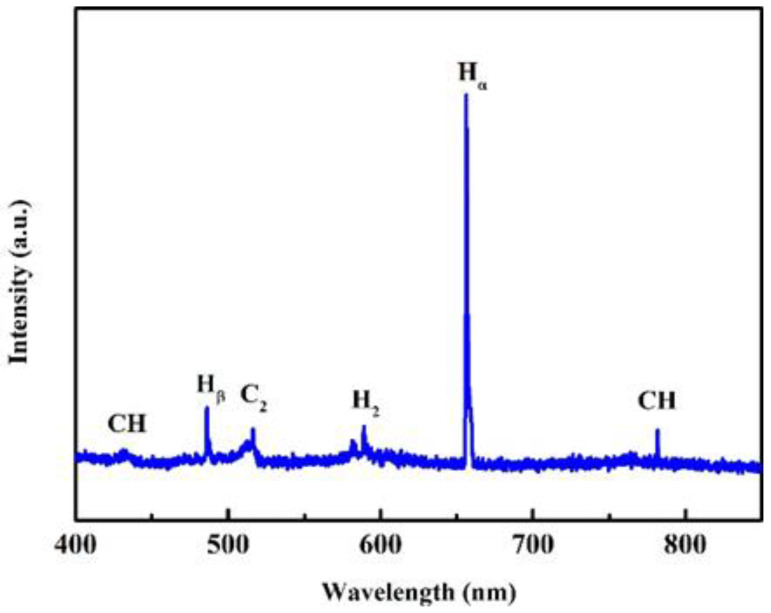
OES spectrum of the growth stage of NNBDD/NDC composite.

**Figure 3 materials-16-06986-f003:**
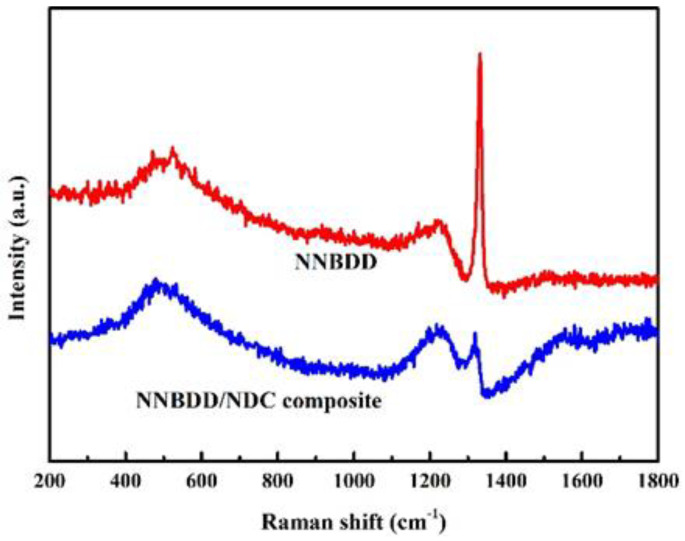
Raman spectra of the NNBDD/NDC composite films (blue line) and NNBDD films (red line).

**Figure 4 materials-16-06986-f004:**
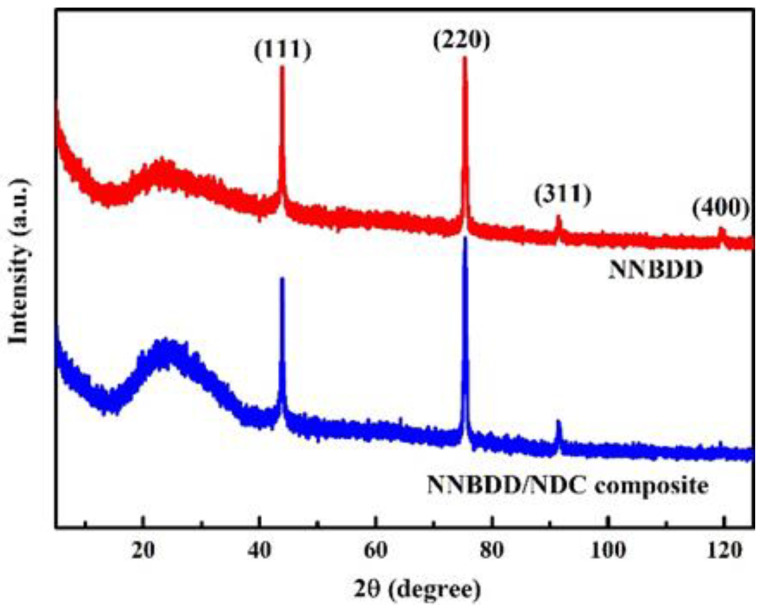
XRD spectra of the NNBDD/NDC composite films (blue line) and NNBDD films (red line).

**Figure 5 materials-16-06986-f005:**
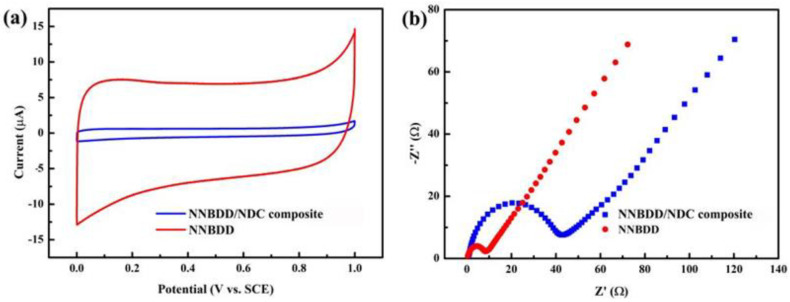
(**a**) CV curves on the NNBDD/NDC composite and NNBDD electrodes in 0.1 M acetate buffer at scan rate of 50 mV s^−1^. (**b**) EIS of NNBDD/NDC composite and NNBDD electrodes tested in a 5 mM Fe(CN)_6_^3−/4−^ solution containing 0.1 M KCl.

**Figure 6 materials-16-06986-f006:**
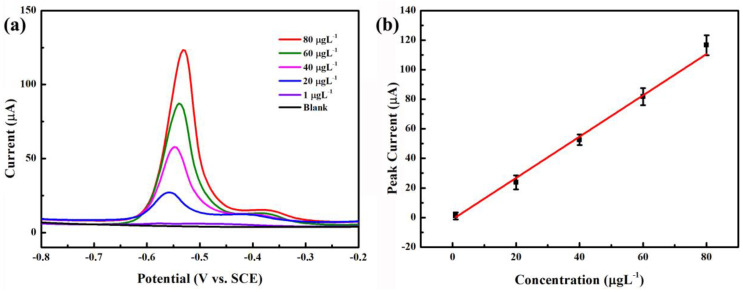
(**a**) The DPASV diagrams of Pb^2+^ with concentrations between 1 and 80 µgL^−1^ on the NNBDD electrode. (**b**) Calibration curve of detecting Pb^2+^. The error bars represent the relative standard deviations of triple measurements. The buffer is 0.1 M acetate buffer (pH = 5.0).

**Figure 7 materials-16-06986-f007:**
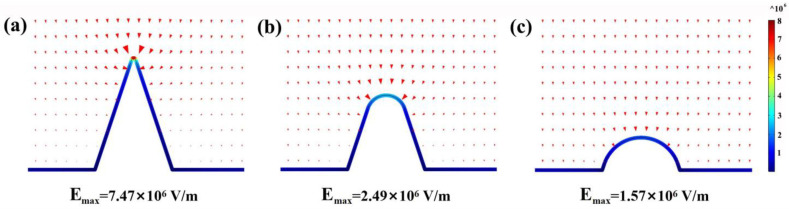
Electric field distributions on the surface of NNBDD at the electrode tip increase as the tip radius decreases. The tip radius of the structure in each panel is (**a**) 5 nm, (**b**) 50 nm, and (**c**) 100 nm.

**Figure 8 materials-16-06986-f008:**
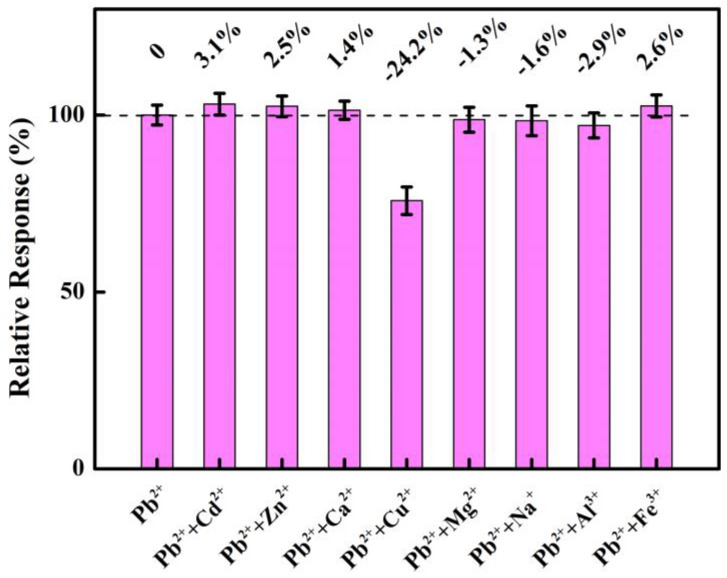
Bar graph of selective ability to resist ion interference.

**Figure 9 materials-16-06986-f009:**
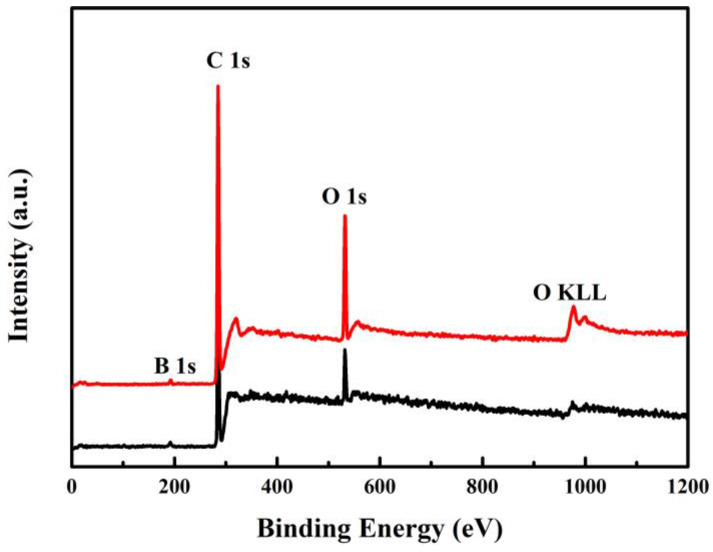
XPS of NNBDD before (black line) and after (red line) electrochemical sensing tests.

**Table 1 materials-16-06986-t001:** Relevant boron doped diamond (BDD) electrodes for Pb^2+^ detection.

Electrodes	Linear Range (µgL^−1^)	Detection Limit (µgL^−1^)	Ref.
BDD	5–1000	3.8	[39]
BDD	20–100	4	[40]
BDD	5–30	2.62	[41]
Nanocrystalline BDD	3.8–45	1.15	[42]
Self-supported BDD	40–600	1.12	[43]
Bi-BDD	1–10	0.51	[44]
Nano-needle BDD	1–80	0.32	This Work

## Data Availability

Not applicable.

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
