# Peer review of "Nano-Needle Boron-Doped Diamond Film with High Electrochemical Performance of Detecting Lead Ions"

_materials, 2023, doi:10.3390/ma16216986_

Round 1

Reviewer 1 Report

Comments and Suggestions for Authors

Author Response

Answers to Reviewers

To Reviewer #1

The authors present a method of fabrication of nanoneedle boron doped diamond and its application for Pb2+ detection. The story is clear and presented. Yet I see several issues to be solved before publication.

The authors thank the Reviewer for offering valuable suggestions and comments to improve the scientific level manuscript. We have carefully considered these comments and suggestions and answered the questions accordingly. The revisions (highlight) have been proposed in the resubmitted manuscript to meet the reviewers' comments.

Q1: The substrates were ultrasonically cleaned, then seeded with nanodiamond powder (supplieris missing…), and then ultrasonically cleaned again. If this is true, I highly doubt that there are any nanodiamonds left on the substrates. Can the authors provide the readers with any microscopical (AFM, SEM) proof, that the substrates are seeded correctly?

Answer: We have provided the readers with SEM and EDS images to prove that the substrates are seeded in supplementary material (Figure S1). Before the deposition of diamond films, the mirror-polished substrates were scratched with nanodiamond powders (about 5 nm) for 30 min on abrasive paper. And then ultrasonicated in an acetone solution with nano-diamond powder for 60 minutes to form nucleation sites. The substrates were ultrasonic cleaned with acetone, ethanol, and purified water for 10 minutes respectively, and dried with nitrogen. The seeds are mostly found in scratches. A small part of the diamond powder remains in the smooth area due to the impact of nanodiamonds on the silicon wafer during ultrasonic action. The scratches and residual nanodiamonds contribute to diamond nucleation. The related illustration is presented in paragraph 5 on page 2.

Q2: Deposition of doped diamond from trimethyl borate is a novel method developed by the group of Kromka and their work (https://doi.org/10.1016/i.diamond.2022.109111) should be cited.

Answer: The paper concerning doped diamond from trimethyl borate by the group of Kromka (Ref. 32 in the revised version) and their related work (Refs. 12, 15, and 45) are cited as suggested.

Q3: Line 129: are the cauliflower clusters really as large as 2-3 mm?

Answer: Thanks for the comments, and all the corrections are presented in the revised version. The size of cauliflower clusters is 2-3 mm.

Q4: I do not agree,that the provided SEM images support the assumption that the needle-curvature is few nm. For that statement, much higher resolution is needed. Therefore, I do not fully approve the label "nanoneedle" in this case at all.

Answer: In the revised version, the insert graph in Figure 1 at much higher resolution has been replaced as suggested.

Q5: Raman spectroscopy: the bands at ~500 and ~1200 are attributed to the Fano effect, yet similar bands are observed on nanodiamonds and are attributed to the presence of sp3 phase (https://doi.org/10.1021/acs.nanolett.1c04887). Can the authors add a discussion on that?

Answer: Thanks for offering valuable suggestions. We have corrected the discussion on the bands at ~500 and ~1200 in Raman spectroscopy as recommended. The asymmetry of the phonon band at 1332 cm-1 is related to the Fano effect. The two broad bands centered at 500 cm-1 and 1200 cm-1 agree with two maxima of phonon density of states of diamond for boron doping. The related illustration is presented in paragraph 1 on page 5.

Q6: In relation to the Fano effect, add the information of the resulting boron doping concentration.

Answer: Thanks for the comments, and the detailed information on the resulting boron doping concentration is added in paragraph 1 on page 5.

Q7: Annealing in air at 800 oC may easily burn diamond. More details on the annealing procedure (e.g.heating/cooling times) are welcome.

Answer: Thanks for your valuable suggestions. And more details on the annealing procedure (e.g.heating/cooling times) are presented in paragraph 1 on page 3.

Reviewer 2 Report

Comments and Suggestions for Authors

The manuscript "Nano-needle boron-doped diamond film with high electro-chemical performance of detecting lead ions" introduced a simple fabrication method of sensing material, consisting of Nano-needle boron-doped diamond. Although the Nano-needle boron-doped diamond demonstrates its good performance, many points are not clear. So, I recommend it for publication in this journal after some minor revisions, which are followed.

1.      It is recommended that authors should check the crystallinity/chemical states of the Nano-needle boron-doped diamond by performing the XRD, and XPS after electrochemical sensing tests.

2.      It is recommended that authors check the electrochemical stability of the Nano-needle boron-doped diamond by performing chronoamperometry.

3.      It is recommended that authors should check the nanostructures of the Nano-needle boron-doped diamond by TEM.

Author Response

Answers to Reviewers

To Reviewer #2:

The manuscript "Nano-needle boron-doped diamond film with high electro-chemical performance of detecting lead ions" introduced a simple fabrication method of sensing material, consisting of Nano-needle boron-doped diamond. Although the Nano-needle boron-doped diamond demonstrates its good performance, many points are not clear. So, I recommend it for publication in this journal after some minor revisions, which are followed.

The authors thank the Reviewer for offering valuable suggestions and comments to improve the scientific level manuscript. We have carefully considered these comments and suggestions and answered the questions accordingly. The revisions (highlight) have been proposed in the resubmitted manuscript to meet the reviewers' comments.

Q1: It is recommended that authors should check the crystallinity/chemical states of the Nano-needle boron-doped diamond by performing the XRD, and XPS after electrochemical sensing tests.

Answer: Thanks for offering valuable suggestions. We added the XRD, and XPS after electrochemical sensing tests to check the crystallinity/chemical states of the Nano-needle boron-doped diamond as recommended. The related illustration is presented in paragraph 4 on page 8.

Q2: It is recommended that authors check the electrochemical stability of the Nano-needle boron-doped diamond by performing chronoamperometry.

Answer: We thank the Reviewer’s suggestion to check the electrochemical stability of the Nano-needle boron-doped diamond by performing chronoamperometry. We have supplemented the chronoamperometry experiments and illustrated them carefully in paragraph 2 on page 7.

Q3: It is recommended that authors should check the nanostructures of the Nano-needle boron-doped diamond by TEM.

Answer: Thanks for the recommendation. We do not yet possess the technology to thin the diamond sample by ion milling to prepare the perfect TEM sample, while we provided the new SEM image at a much higher resolution in Figure 1 to check the nanostructures of the Nano-needle boron-doped diamond.

Reviewer 3 Report

Comments and Suggestions for Authors

This is a simple paper on how to measure lead ions using nano-needle boron-doped diamond. However, I would like to suggest a few things to improve the paper.

What is the ratio of NNBDD to NDC contained in NNBDD/NDC used as a control group? Was the same NNBDD/NDC used in all experiments? Does the ratio of NNBDD and NDC included in NNBDD/NDC have any effect on performance?

COMSOL Multiphysics was used as key evidence supporting the performance of NNBDD. However, there is no information about which modules and methodologies of COMSOL Multiphysics were used for the analysis. Extensive detail needs to be added in this area.

Comments on the Quality of English Language

No comment.

Author Response

Answers to Reviewers

To Reviewer #3:

This is a simple paper on how to measure lead ions using nano-needle boron-doped diamond. However, I would like to suggest a few things to improve the paper.

The authors thank the Reviewer for offering valuable suggestions and comments to improve the scientific level manuscript. We have carefully considered these comments and suggestions and answered the questions accordingly. The revisions (highlight) have been proposed in the resubmitted manuscript to meet the reviewers' comments.

Q1: What is the ratio of NNBDD to NDC contained in NNBDD/NDC used as a control group?

Answer: The ratio of NNBDD to NDC is close to 1:1, which depends on the weight loss of the NNBDD/NDC composite film during heating in air.

Q2: Was the same NNBDD/NDC used in all experiments?

Answer: Yes, the same NNBDD/NDC was used in all experiments.

Q3: Does the ratio of NNBDD and NDC included in NNBDD/NDC have any effect on performance?

Answer: So far, we are not very clear about the mechanism of adjusting the ratio of NNBDD to NDC. Our initial idea is to control the ratio of NNBDD to NDC by changing the ratio of methane to hydrogen during the growth. However the experimental results are not consistent with our expected ideas. NNBDD was formed after annealing under conditions with high methane ratio growth (the CH4/H2/B flow rate was 20/200/2 sccm). While porous diamond was formed after annealing under condition with lower methane ratio growth (the CH4/H2/B flow rate was 10/200/2 sccm), and the nanoneedle-like diamond disappeared. Diamonds cannot be formed with a very high methane ratio (the CH4/H2/B flow rate was 30/200/2 sccm), and the graphite substances can be formed due to the incomplete etching of NDC by hydrogen plasma. The NNBDD/NDC composite only appears at appropriate methane ratios. Therefore, the ratio of NNBDD and NDC in the NNBDD/NDC composite is fixed under the same experimental conditions.

    We think the ratio of NNBDD to NDC would have some effect on performance if the ratio could be adjusted because the ratio of NNBDD and NDC might affect the density of the nanoneedle. Thank you for your valuable suggestions very much. In the following systematic research, we will explore systematic research on possible factors such as pressure, temperature, power, and other parameters during the growth, which might affect ratio of NNBDD and NDC, the morphology and performance of NNBDD. It will be an interesting work that more time is needed to confirm the controllable ratio of NNBDD to NDC experimentally.

Q4: COMSOL Multiphysics was used as key evidence supporting the performance of NNBDD. However, there is no information about which modules and methodologies of COMSOL Multiphysics were used for the analysis. Extensive detail needs to be added in this area.

Answer: Thanks for offering valuable suggestions. We have supplemented the modules and methodologies of COMSOL Multiphysics. The related illustration is presented in paragraph 3 on page 3.

Round 2

Reviewer 3 Report

Comments and Suggestions for Authors

This version of the manuscript is acceptable for publication.